# Gut Microbial Dysbiosis Associated with Type 2 Diabetes Aggravates Acute Ischemic Stroke

Xiaojiao Chen,[a] Qiheng Wu,[b] Xuxuan Gao,[a] Huidi Wang,[b] Jiajia Zhu,[b] Genghong Xia,[b] Yan He,[a] Wei Song,[b] Kaiyu Xu[a]

[a]Microbiome Medicine Center, Division of Laboratory Medicine, Zhujiang Hospital, Southern Medical University, Guangzhou, China
[b]Department of Neurology, Nanfang Hospital, Southern Medical University, Guangzhou, China

Xiaojiao Chen, Qiheng Wu, and Xuxuan Gao contributed equally to this work. Author order was determined on the basis of seniority.

**ABSTRACT** Type 2 diabetes (T2D) is an independent risk factor for acute ischemic stroke (AIS), but the underlying mechanisms remain elusive. Because the gut microbiota plays a causal role in both T2D and AIS, we wondered whether gut dysbiosis in T2D aggravates stroke progression. We recruited 35 T2D, 90 AIS, 60 AIS with T2D (AIS_T2D) patients, and 55 healthy controls and found that AIS and T2D had an additive effect on AIS_T2D patient gut dysbiosis by exhibiting the largest difference from the heathy controls. In addition, we found that the degree of gut dysbiosis associated with T2D was positively correlated with the National Institutes of Health Stroke Scale (NIHSS), modified Rankin score (mRS), and Essen stroke risk score in patients with AIS, including AIS and AIS_T2D patients. Compared with mice colonized with gut microbiota from healthy controls poststroke modeling, germ-free (GF) mice colonized with gut microbiota from T2D patients showed exacerbated cerebral injury and impaired gut barrier function. Specifically, exacerbated brain injury and gut barrier dysfunction in T2D-treated GF mice were significantly associated with a reduction in short-chain fatty acid (SCFA)-producing bacteria. Our study showed that T2D and AIS have an additive effect on AIS_T2D patient gut microbiota dysbiosis. T2D-associated gut microbiota dysbiosis is associated with stroke severity in AIS patients and aggravates stroke progression in mice.

**IMPORTANCE** We demonstrated an additive effect of type 2 diabetes (T2D) and acute ischemic stroke (AIS) on AIS with T2D (AIS_T2D) patient gut microbiota dysbiosis, and gut dysbiosis associated with T2D was positively correlated with stroke severity in AIS patients. Through animal experiments, we found that cerebral injury was exacerbated by fecal microbiota transplantation from T2D patients compared with that from healthy controls, which was associated with a reduction in short-chain fatty acid (SCFA)-producing bacteria. This study provided a novel view that links T2D and AIS through gut microbial dysbiosis.

**KEYWORDS** gut microbiota, acute ischemic stroke, type 2 diabetes

Address correspondence to Wei Song, songwei75@smu.edu.cn, or Kaiyu Xu, xukaiyu09@smu.edu.cn.

The authors declare no conflict of interest.

Stroke is one of the most common causes of death and disability worldwide and is the leading cause of death and disability-adjusted life-years in China (1, 2). Among many modifiable risk factors, type 2 diabetes mellitus (T2D) is an independent risk factor for the progression of ischemic stroke (3). Individuals with T2D have a 1.5 to 3 times higher risk of stroke than nondiabetic individuals (4), and diabetic patients experience more adverse events and subsequent mortality after stroke (5). According to a study in the United Kingdom, the hazard ratio for ischemic stroke in diabetic individuals is 2.3 (95% CI 2.0 to 2.7) compared with that in nondiabetic individuals (6).

For metabolic disorders such as T2D and cardio-cerebrovascular diseases such as stroke, there is a newly identified common risk factor, the gut microbiome (7, 8). Studies of people from different ethnicities have found that individuals with T2D showed reduced

**TABLE 1** Characteristics of the healthy controls, AIS patients, T2D patients, and AIS_T2D patients[a]

| Parameter | Value for: | | | |
| --- | --- | --- | --- | --- |
| | Healthy controls | AIS patients | T2D patients | AIS_T2D patients |
| No. | 55 | 90 | 35 | 60 |
| Age | 61 (4) | 61 (18) | 65 (11) | 63 (16) |
| Sex, M | 0.5636 | 0.6556 | 0.6571 | 0.6500 |
| BMI, kg/m$^2$ | 23.00 (5.00) | 23.66 (3.94) | 24.00 (3.00)* | 23.87 (2.88) |
| WBC, 10$^9$/L | 5.86 (1.79) | 7.49 (3.15)*** | 7.04 (2.67)* | 8.22 (4.16)*** |
| LYM, 10$^9$/L | 2.01 (0.83) | 1.83 (0.69) | 1.90 (0.63) | 2.02 (0.92) |
| NEU, 10$^9$/L | 3.18 (1.11) | 4.65 (2.59)*** | 3.38 (2.08) | 5.13 (3.33)*** |
| MONO, 10$^9$/L | 0.33 (0.11) | 0.47 (0.19)*** | 0.35 (0.24) | 0.47 (0.20)*** |
| RBC, 10$^{12}$/L | 4.71 (0.59) | 4.78 (0.74) | 5.09 (0.92)* | 4.91 (0.79) |
| HGB, g/L | 143.00 (23.00) | 140.00 (18.25) | 147.00 (28.00) | 141.50 (27.50) |
| PLT, 10$^9$/L | 234 (86) | 228.50 (73.50) | 196 (68)* | 217 (93.00) |
| GLU, mmol/L | 4.84 (0.71) | 5.05 (1.32) | 6.81 (3.24)*** | 8.60 (3.96)*** |
| BUN, mmol/L | 4.90 (1.77) | 5.00 (2.41) | 5.41 (1.55) | 5.48 (1.80) |
| Cr, $\mu$mol/L | 66 (18) | 79.00 (28.00)*** | 69 (18.25) | 73 (40.75) |
| UA, $\mu$mol/L | 333 (100) | 368 (112.00) | 350 (102) | 346 (130.25) |
| TP, g/L | 68.90 (5.20) | 64.30 (7.00)*** | 71.90 (4.20)** | 65.90 (7.75)** |
| ALB, g/L | 42.30 (3.80) | 38.35 (5.25)*** | 42.10 (1.60) | 39.30 (5.25)*** |
| GLB, g/L | 26.40 (5.00) | 25.70 (5.07) | 29.80 (2.10)** | 26.30 (4.70) |
| SBP, mmHg | 124 (18) | 142 (25)*** | 132 (33)* | 143 (36)*** |
| DBP, mmHg | 72 (15) | 84 (21)*** | 73 (20) | 82 (14)*** |
| TC, mmol/L | 4.94 (0.82) | 4.53 (1.750) | 4.99 (1.88) | 4.78 (1.415) |
| LDL, mmol/L | 3.05 (0.66) | 2.91 (1.33) | 2.98 (1.59) | 3.11 (1.150) |
| HDL, mmol/L | 1.23 (0.34) | 0.95 (0.320)*** | 1.18 (0.27) | 0.91 (0.305)*** |
| VLDL, mmol/L | 0.58 (0.28) | 0.62 (0.415) | 0.66 (0.45) | 0.70 (0.495)** |
| TG, mmol/L | 1.06 (0.59) | 1.22 (0.840)** | 1.26 (0.71)* | 1.84 (1.025)*** |

[a]For all records except number and gender, data represent the median (interquartile range). For all records except gender, the P values represent the results of the Wilcoxon rank sum test. The P value of gender represents the result of Pearson's chi-square test. BMI, body mass index; WBC, white blood cell count; LYM, lymphocytes; NEU, neutrophils; MONO, monocytes; RBC, red blood cells; HGB, hemoglobin; PLT, platelets; GLU, glucose; BUN, blood urea nitrogen; Cr, creatinine; UA, uric acid; TP, total protein; ALB, albumin; GLB, globulin; SBP, systolic blood pressure; DBP, diastolic blood pressure; TC, total cholesterol; LDL, low-density lipoprotein; HDL, high-density lipoprotein; VLDL, very low-density lipoprotein; TG, triglycerides. *, $P < 0.05$; **, $P < 0.01$; ***, $P < 0.001$ versus healthy controls.

butyrate-producing bacteria leading to gut inflammation (9). A preclinical trial study confirmed that microbiota transplantation from lean donors improved insulin sensitivity in male recipients with metabolic syndrome, suggesting a causal connection between T2D and gut microbiota (10, 11). In addition, gut microbiota and its metabolites are also associated with stroke, i.e., ischemic stroke causes gut microbiota dysbiosis, and such dysbiosis may exacerbate brain infarction poststroke (8, 12). The microbiome-derived metabolite trimethylamine-N-oxide contributes to the progression of atherosclerosis (13, 14).

Because T2D is an independent risk factor for stroke, it is very intriguing to explore the role of the gut microbiome in this correlation. Here, we hypothesized that T2D aggravates stroke progression by inducing gut microbiota dysbiosis. We compared the similarities and differences in the gut microbiome in acute ischemic stroke (AIS) patients, T2D patients, and AIS patients complicated with T2D (AIS_T2D). In addition, we evaluated the role of gut microbiota dysbiosis associated with T2D in aggravating stroke progression.

## RESULTS

**AIS and T2D had an additive effect on gut microbiota dysbiosis in AIS_T2D patients.** Our study recruited 90 AIS patients, 60 AIS_T2D patients, 35 T2D patients, and 55 healthy controls. The characteristics of the study participants are included in Table 1. For all individuals involved in the present study, most of their gut bacteria fell into the phyla Bacteroidetes, Firmicutes, Proteobacteria, and Fusobacteria, and at the genus level, *Bacteroides*, *Prevotella*, *Faecalibacterium*, and *Parabacteroides* occupied the highest abundance (Fig. 1A and B).

Specifically, compared with healthy controls, AIS patients and T2D patients exhibited similar differences in gut microbial alteration patterns. Principal coordinate analysis (PCoA) based on unweighted UniFrac distance showed that the spatial distributions of both AIS

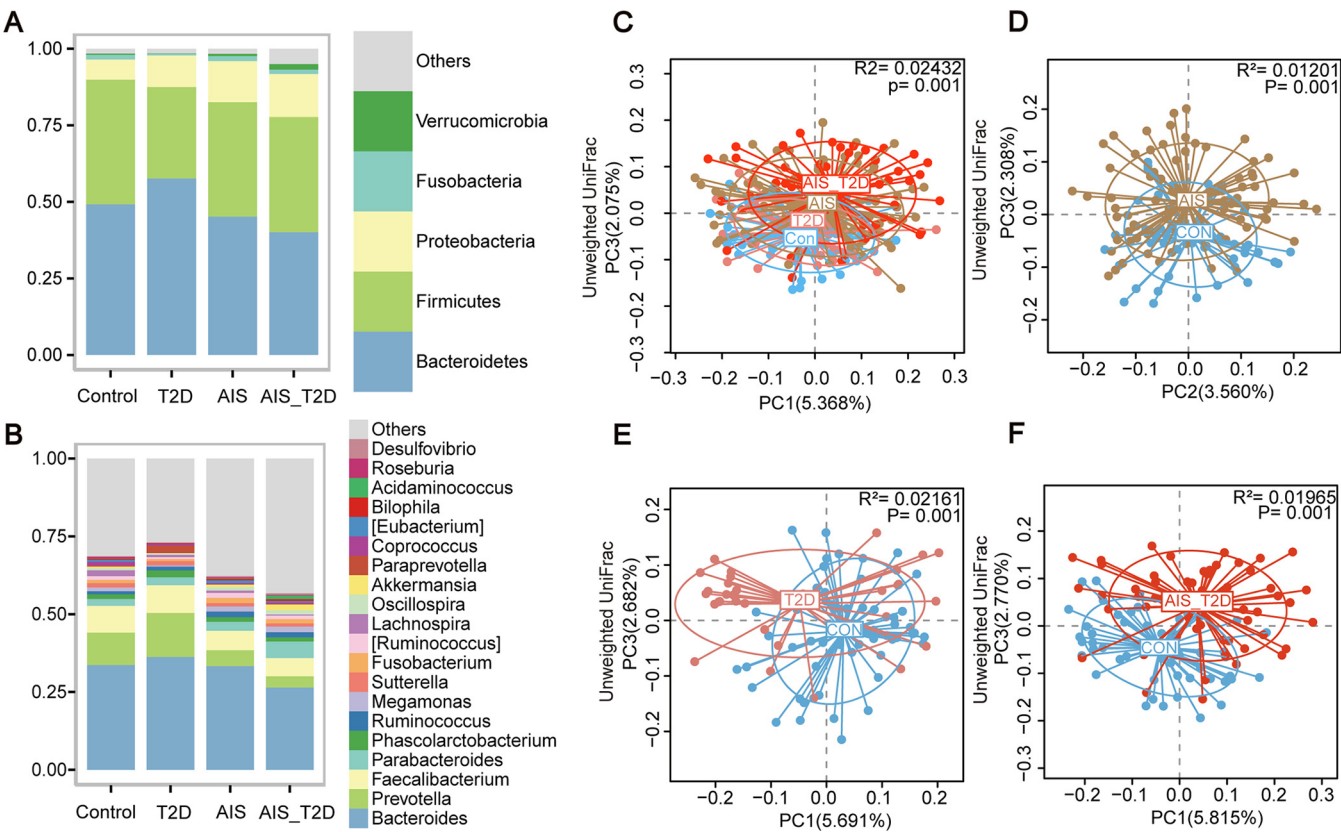

**FIG 1** Gut microbiota composition among the AIS (*n* = 86), T2D (*n* = 35), AIS_T2D (*n* = 59), and control groups (*n* = 55). Average relative abundances of the predominant bacterial taxa at the phylum (A) and genus (B) levels. The overall PCoA showing the gut microbial composition pattern of the AIS, T2D, AIS_T2D, and control groups (C). Obvious dysbiosis of the gut microbiome in unweighted UniFrac-based PCoA plots of AIS patients (D), T2D patients (E), and AIS_T2D patients (F) compared with the control group. AIS, acute ischemic stroke; PCoA, principal coordinate analysis.

patients and T2D patients were significantly different from those of healthy controls (Fig. 1C to E). Linear discriminant analysis effect size (LEfSe) analysis revealed that AIS patients showed an increased abundance of Proteobacteria, Gammaproteobacteria, Enterobacteriaceae, *Deltaproteobacteria*, and Desulfovibrionaceae. In contrast, *Prevotella*, *Faecalibacterium*, *Lachnospira*, and *Coprococcus* were decreased in AIS patients (Fig. S1A). Compared with healthy controls, Enterobacteriaceae, Lactobacillales, Lactobacillaceae, and *Lactobacillus* were significantly enriched, while *Lachnospira* was decreased in T2D patients (Fig. S1B). Furthermore, AIS_T2D patients showed a clear separation from healthy controls in PCoA (Fig. 1F). Compared with those in healthy controls, Enterobacteriaceae and Desulfovibrionaceae were significantly enriched, while *Bacteroides*, *Faecalibacterium*, *Lachnospira*, and *Coprococcus* were significantly depleted in AIS_T2D patients (Fig. S1C). Together, AIS and T2D have an additive effect on gut microbiota dysbiosis in AIS_T2D patients.

The gut microbiota configurations of AIS or T2D patients exhibited alterations in the same direction as healthy controls, while AIS_T2D patients showed a greater distance from healthy controls than the AIS and T2D groups did in the same direction, according to ordination analysis (Fig. 1C and Fig. 2A). The unweighted UniFrac distance revealed that AIS_T2D patients exhibited a significantly greater distance than both AIS patients and T2D patients (Fig. 2B). To further explore the gut microbial alteration patterns among AIS, T2D, and AIS_T2D patients, we further analyzed the abundance of selected bacteria according to LEfSe analysis and found similar microbial alteration patterns among the three patient groups. AIS_T2D patients showed the highest abundance of Enterobacteriaceae, *Lactobacillus*, and *Megasphaera* and the lowest abundance of *Lachnospira* and *Coprococcus* (Fig. 2C). T2D patients and AIS_T2D patients both exhibited a lower abundance of *Roseburia* than healthy controls. T2D patients exhibited the lowest abundance of Dorea among the four groups of people.

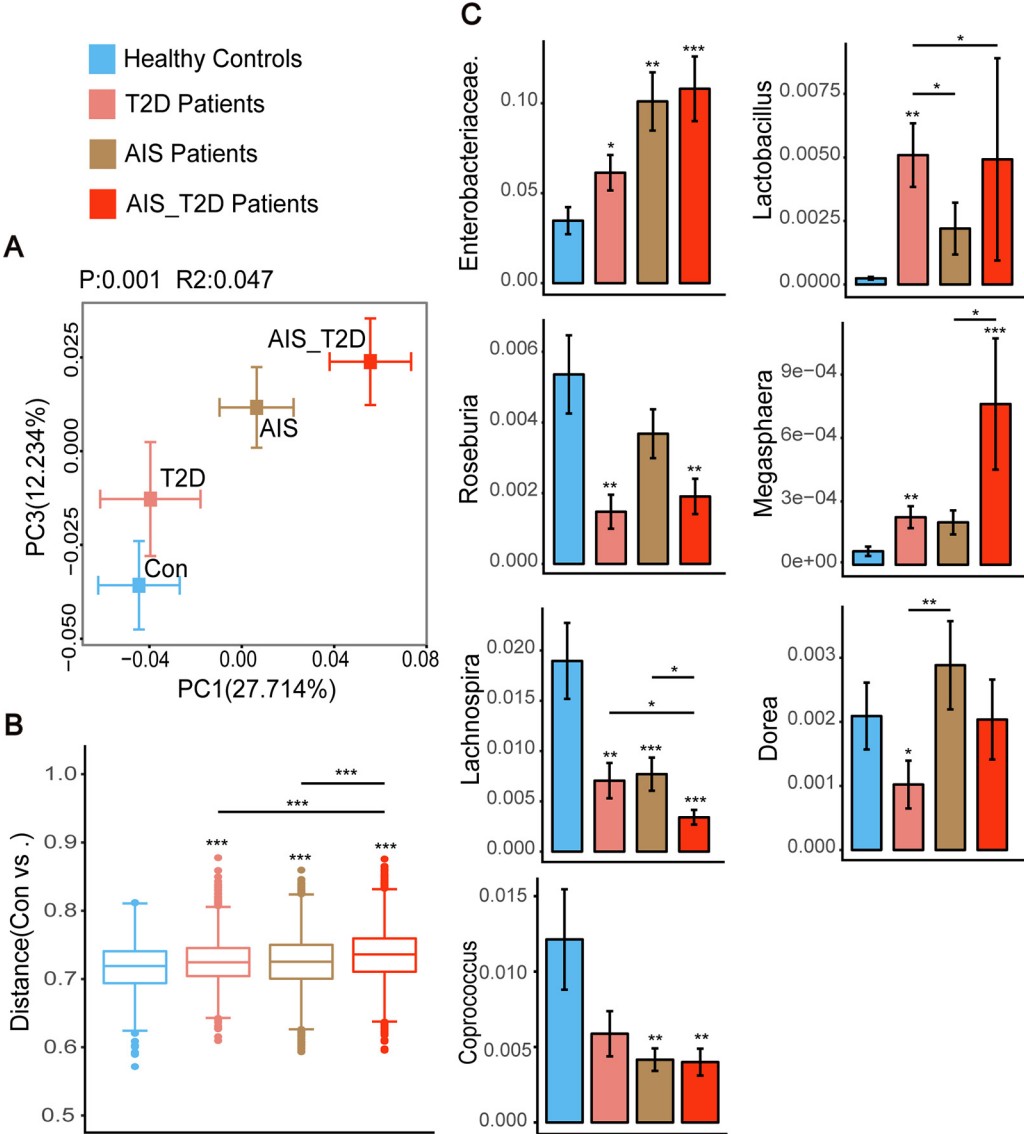

**FIG 2** AIS and T2D had an additive effect on the gut microbiota composition of AIS_T2D patients. (A) Different microbial patterns were exhibited in the PCoA based on weighted UniFrac distances. (B) Box plot illustrating the unweighted UniFrac distances of different patient groups to controls. (C) The relative abundance of selected bacteria using the Wilcoxon rank sum test. *, $P < 0.05$; **, $P < 0.01$; and ***, $P < 0.001$ based on the Wilcoxon rank sum test. AIS, acute ischemic stroke; PCoA, principal coordinate analysis.

**The T2D gut microbial dysbiosis index was significantly positively correlated with the severity and prognosis of patients with AIS.** To further explore this additive effect on the gut microbiota, we established a gut microbial index to represent the overall gut dysbiosis associated with T2D. This index was formulated by the accumulated abundance of positively and negatively T2D-associated operational taxonomic units (OTUs) valued by their significance. We found that this index could reliably reflect gut dysbiosis in T2D with significantly higher indices in T2D patients than in healthy controls (Fig. 3A). This index also reflected gut dysbiosis in AIS_T2D patients with significantly higher indices than in AIS patients (Fig. 3B).

To explore whether this T2D gut microbial index was correlated with the severity and prognosis of patients with AIS, including AIS and AIS_T2D patients, we conducted Spearman correlation analysis between the T2D index and the neurological function scores, including National Institutes of Health Stroke Scale (NIHSS) on admission and modified Rankin score (mRS) and Essen stroke risk scores at chronic stage. We found

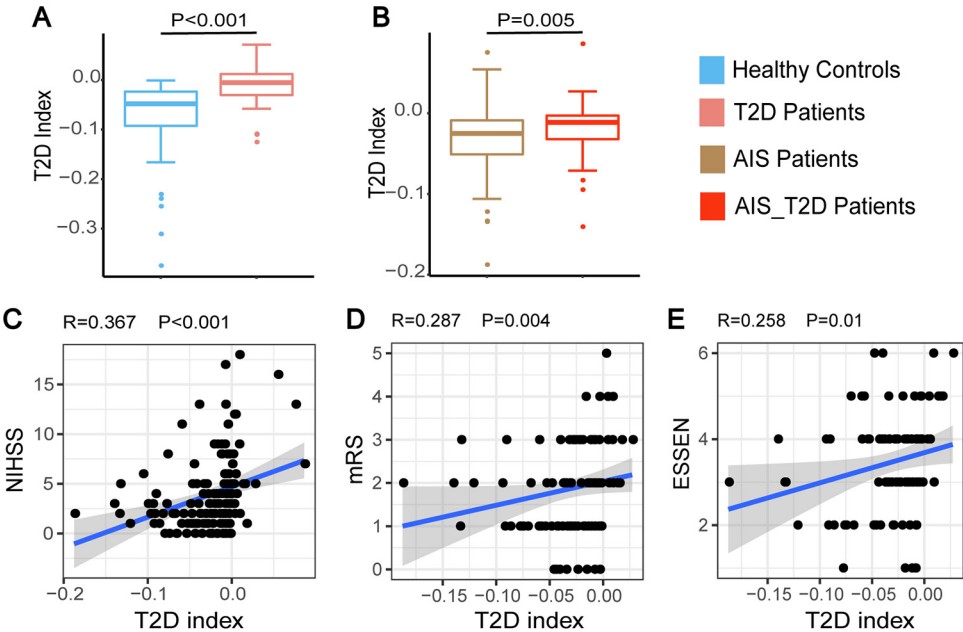

**FIG 3** The T2D gut microbial index was significantly associated with the severity and prognosis of patients with AIS. (A–B) The T2D gut microbial index significantly distinguished between T2D patients and healthy controls (A) and between AIS_T2D patients and AIS patients (B). (C–E) The T2D gut microbial index was significantly positively correlated with the NIHSS score (C), Essen stroke risk score (D), and mRS score (E) in patients with AIS. AIS, acute ischemic stroke; NIHSS, National Institutes of Health Stroke Scale; mRS, modified Rankin score.

that the NIHSS score, mRS score, and Essen stroke risk score were significantly positively correlated with the T2D gut microbial index (Fig. 3C to E), indicating that gut dysbiosis evaluated by the T2D gut microbial index was associated with the severity and prognosis of stroke patients.

**Transplantation of fecal samples from T2D patients exacerbated brain infarction poststroke.** To investigate whether gut dysbiosis associated with T2D contributes to brain injury or neurological deficits after stroke, we transplanted fecal samples from T2D patients and healthy controls to germfree (GF) mice and subsequently conducted middle cerebral artery occlusion (MCAO) (Fig. 4A). T2D-treated GF mice exhibited significantly larger brain infarction and a higher modified neurological severity score (mNSS) than the control group (Fig. 4B to D). In addition, gut microbiota associated with T2D impaired the gut barrier in GF mice. Serum lipopolysaccharide (LPS) and LPS-binding protein (LBP) levels were significantly increased in T2D-treated GF mice (Fig. 4E). Relative expression levels of tight junction proteins, including *Tjp1*, *Ocln*, and *Cldn4*, in the colon and *Ocln* and *Cldn4* in the ileum were significantly decreased in T2D-treated GF mice (Fig. 4F). Immunofluorescence revealed that ZO-1 and occludin were abnormally distributed along the epithelial sheet, and the staining for both proteins was weaker and more disrupted in ileum and colon tissues in T2D-treated GF mice than in the control group (Fig. 4G). Furthermore, proinflammatory cytokines such as *Cxcl1* and *Cxcl2* were significantly increased in T2D-treated GF mice compared with those in the control group (Fig. 4F). Together, fecal microbiota transplantation (FMT) from T2D individuals exacerbated cerebral injury and impaired gut barrier function after stroke.

**Short-chain fatty acid (SCFA)-producing bacteria are significantly negatively associated with stroke progression in T2D-treated GF mice.** To explore which OTU was associated with brain injury poststroke in T2D-treated GF mice, correlation analysis was performed. As shown in Fig. 5, a total of 61 OTUs were significantly correlated with mouse outcome indices. Twenty-two OTUs, namely, two OTUs from *Lachnospira*, 13 OTUs from Ruminococcaceae, three OTUs from Lachnospiraceae, OTU584978 from *Ruminococcus*, OTU289306 from Oscillospira, OTU537538 from Clostridiales, and OTU214919 from *Turicibacter*, were significantly negatively correlated with exacerbated brain infarction, neurological deficits and gut barrier dysfunction in T2D-treated GF mice. In contrast, a total of 16

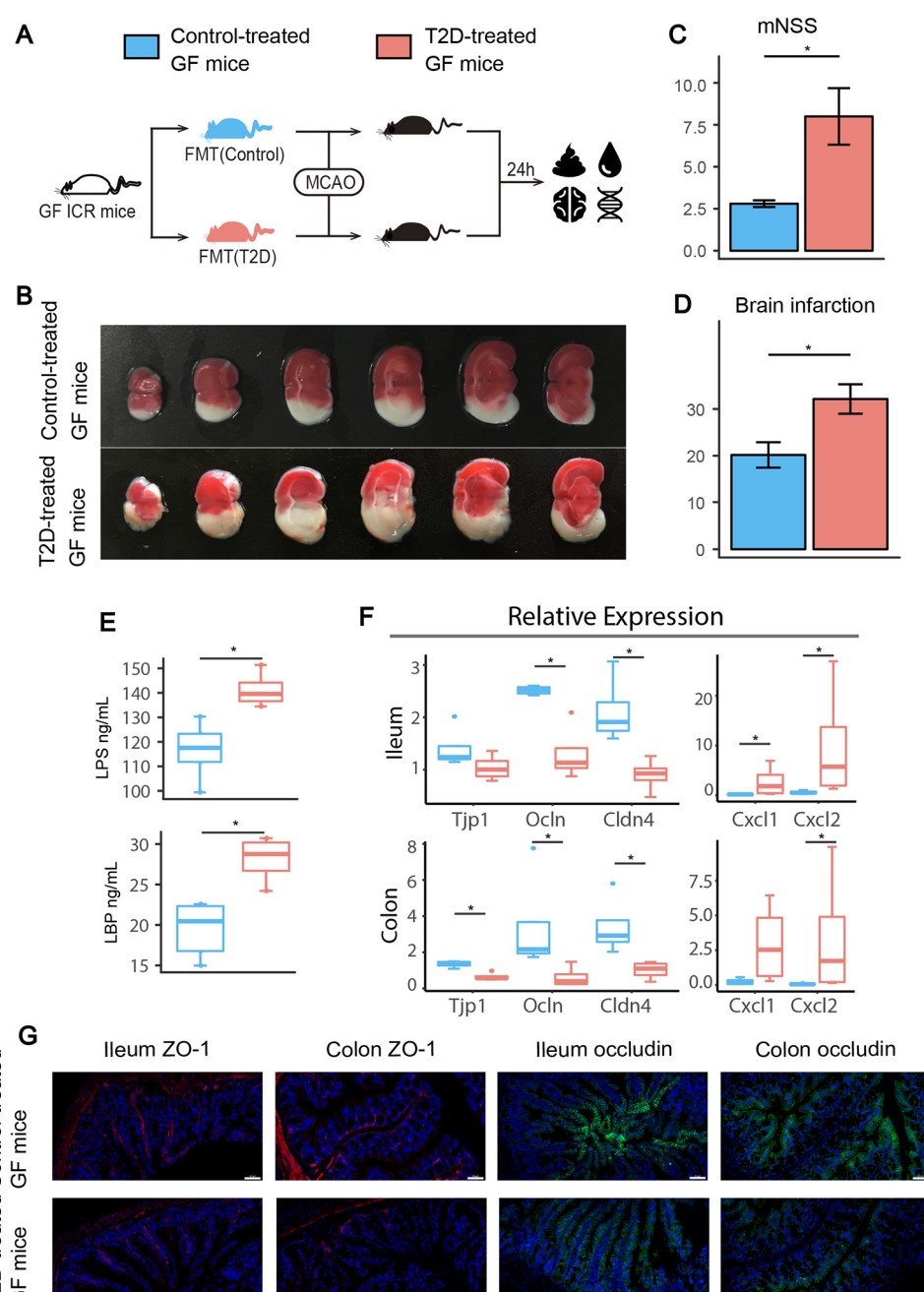

**FIG 4** T2D-treated GF mice exhibited exacerbated cerebral injury and gut barrier dysfunction. (A) Experimental design of FMT and MCAO surgery in GF mice. (B) Representative images of TTC-stained coronal brain sections. (C–D) Comparison of the brain infarction ratio and mNSS between GF mice transplanted with fecal samples from healthy controls (*n* = 5, blue) and T2D patients (*n* = 4, red). (E) Serum LBP and LPS levels in control and T2D-treated GF mice. (F) Relative expression of the *Tjp1*, *Ocln*, *Cldn4*, *Cxcl1*, and *Cxcl2* genes and (G) immunofluorescent staining for occludin and ZO-1 (magnification, ×200) in mouse ileum and colon tissues. Scale bar 50 μm. *, *P* < 0.05 and **, *P* < 0.01 based on the Wilcoxon rank sum test. FMT, fecal microbiota transplantation; MCAO, middle cerebral artery occlusion; mNSS, modified neurological severity score.

OTUs, namely, OTU1145098 from *Ruminococcus gauvreauii*, OTU180731 from Parabacteroides, three OTUs from Clostridiales, two OTUs from Coprococcus, two OTUs from *Parabacteroides distasonis*, two OTUs from Lachnospiraceae, OTU264325 from Rikenellaceae, OTU3231096 from S24-7, OTU4331723 from Ruminococcus, OTU334267 from Ruminococcaceae, and OTU586453 from Christensenellaceae, showed significantly positive correlations with brain injury and gut barrier dysfunction in T2D-treated GF mice. Two controversial OTUs

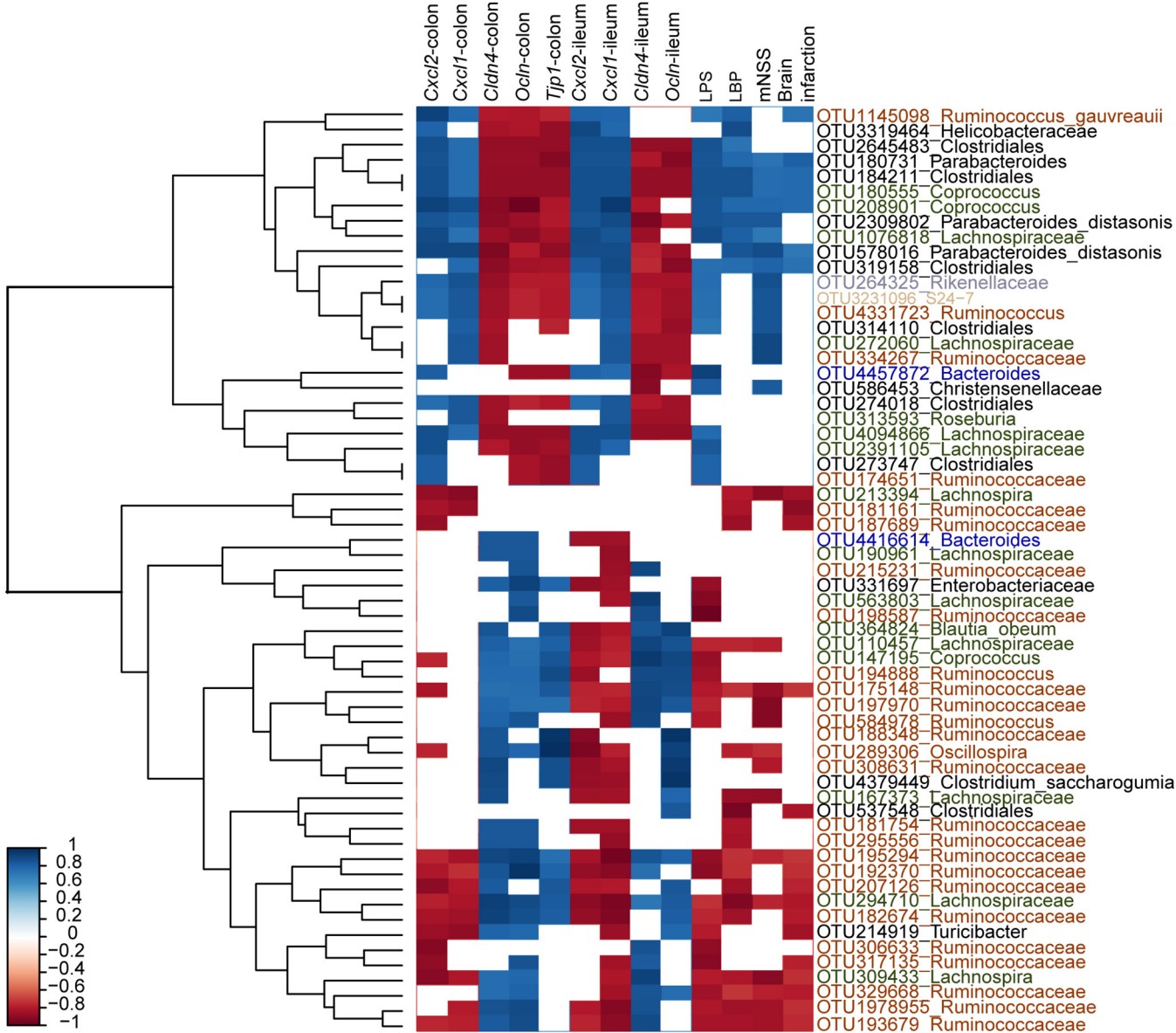

**FIG 5** Heatmap demonstrating the important OTUs and their correlation relationships with brain injury and gut barrier dysfunction in GF mice. A total of 61 OTUs were significantly correlated with the above indices. The boxes shown in red are negatively correlated, and the boxes in blue are positively correlated. The magnitude of the correlation is illustrated by the intensity of the color. OTUs, operational taxonomic units.

in Bacteroides showed the difference. OTU4416614 was positively correlated with gut barrier dysfunction, and OTU4457872 showed the opposite correlation.

Based on the correlation results, OTUs that showed remarkable correlations with brain infarction, neurological deficits or gut barrier dysfunction were mainly from Ruminococcaceae, *Lachnospira*, and S24-7. Specifically, SCFA-producing bacteria, such as Ruminococcaceae and *Lachnospira*, showed a negative correlation with stroke progression and gut barrier dysfunction in T2D-treated GF mice (Table 2). Mean while, S24-7, an LPS-producing bacterium, showed a positive correlation. Collectively, SCFA-producing bacteria were negatively associated with stroke progression and gut barrier dysfunction.

We performed an additional KEGG functional analysis using Phylogenetic Investigation of Communities by Reconstruction of Unobserved States (PICRUSt) based on 16S rRNA gene sequences. The two groups of GF-treated mice formed significantly different clusters in the principal-component analysis (PCA) plot (Fig. S2A). A total of 78.4% of the variation was captured by the first principal component (PC1). Among the 6,916 affiliated KEGG pathways,

**TABLE 2** Spearman's correlation for different bacteria and mouse injury indices after MCAO

| Bacteria | Target | R | P value |
|---|---|---|---|
| Ruminococcaceae | Cerebral infarction ratio | −0.900 | 0.002 |
| | mNSS | −0.825 | 0.000 |
| | LBP | −0.800 | 0.014 |
| | LPS | −0.767 | 0.021 |
| | *Ocln* in ileum | 0.738 | 0.046 |
| | *Cldn4* in ileum | 0.833 | 0.015 |
| | *Cxcl1* in ileum | −0.810 | 0.022 |
| | *Ocln* in colon | 0.762 | 0.037 |
| | *Cldn4* in colon | 0.738 | 0.046 |
| | *Cxcl1* in colon | −0.786 | 0.028 |
| | *Cxcl2* in colon | −0.786 | 0.028 |
| *Lachnospira* | Cerebral infarction ratio | −0.683 | 0.050 |
| | mNSS | −0.908 | 0.000 |
| | LBP | −0.700 | 0.043 |
| | LPS | −0.750 | 0.026 |
| | *Cldn4* in ileum | 0.905 | 0.005 |
| | *Cxcl1* in ileum | −0.738 | 0.046 |
| | *Cxcl1* in colon | −0.762 | 0.037 |
| | *Cxcl2* in colon | −0.833 | 0.015 |
| *S24-7* | Cerebral infarction ratio | 0.740 | 0.029 |
| | mNSS | 0.638 | 0.073 |
| | LBP | 0.773 | 0.019 |
| | LPS | 0.798 | 0.013 |
| | *Ocln* in ileum | −0.723 | 0.045 |
| | *Cldn4* in ileum | −0.771 | 0.027 |
| | *Ocln* in colon | −0.747 | 0.035 |

1,591 were shown to achieve a statistically significant difference at $P < 0.05$, and among them, 22 are shown in Fig. S2B based on $P < 10^{-5}$. Notably, significant elevation of glycerol uptake facilitator protein and peroxiredoxin and a decrease in putative tributyrin esterase were observed in T2D-treated GF mice compared with control-treated GF mice according to the KEGG Orthologs (KO) database (Fig. S2B).

## DISCUSSION

Diabetes patients usually experience increased adverse events and subsequent mortality after stroke, which is a growing challenge for health care systems (15). In this study, the gut microbial composition of AIS patients resembled that of T2D patients. Both AIS patients and T2D patients revealed a remarkable alteration in gut microbial composition by exhibiting a decreased abundance of commensal or beneficial bacteria, including *Prevotella*, *Faecalibacterium*, *Lachnospira*, and *Coprococcus*, in accordance with previous studies (12, 16). *Faecalibacterium*, *Lachnospira*, and *Coprococcus* are known to be SCFA-producing bacteria. SCFAs play critical roles at both the intestinal and extraintestinal levels, including maintaining intestinal barrier functions, reducing the severity of mucosal inflammation, promoting intestinal oxidative status, enhancing colon motility functions, and improving many pathologies (17, 18). In addition, SCFAs also help to improve metabolic functions in T2D by inhibiting inflammation and insulin resistance, augmenting GLP-1 secretion, and ameliorating $\beta$-cell function (19). On the other hand, opportunistic pathogens, such as Proteobacteria and Enterobacteriaceae, showed a growth advantage in both AIS patients and T2D patients. Proteobacteria is proposed as a marker for dysbiosis of the microbial community and a potential diagnostic criterion for metabolic and immune disorders (20) and is also enriched in hosts with glucose intolerance utilizing artificial sweeteners and dietary emulsifiers (21). Enterobacteriaceae is regarded as a major harmful member of the human symbiotic microbial community, triggering enteropathy and interfering with the development of mucosal immunity and healthy gut microbes (22). Enterobacteriaceae has also been identified as an independent risk factor in predicting early-stage recovery for AIS patients (8).

AIS_T2D patients showed the highest extent of gut microbial dysbiosis with increased abundance of opportunistic pathogens belonging to the phylum Proteobacteria and reduced commensal or beneficial bacteria, implying that AIS and T2D had an additive effect on gut microbiota dysbiosis in AIS_T2D patients. Previous studies suggested that the additive effects of the gut microbiota were mediated by gut dysbiosis, diet, and lifestyle changes (23, 24). Here, we identified an additive effect contributed by diseases, such as AIS and T2D, and superficially addressed this association, requiring future studies to investigate the underlying mechanism.

We emphasize an important role for gut dysbiosis associated with T2D in exacerbating the severity and prognosis of stroke. By formulating a T2D gut microbial dysbiosis index, we found that T2D gut microbial indices showed a significantly positive correlation with the NIHSS, Essen stroke risk score, and mRS score, supporting an important role of gut dysbiosis as a risk factor in predicting the severity and poor outcomes of stroke patients. Similarly, an index formulated based on gut dysbiosis of stroke patients was also significantly associated with brain injury (25). By conducting FMT of T2D-associated gut microbiota on GF mice, we found that gut dysbiosis associated with T2D may exacerbate the aggregation of brain injury and neurological defects poststroke. In addition, FMT of T2D-associated gut microbiota induces gut barrier dysfunction in GF mice, including elevated serum LBP and LPS levels and impairment of tight junction protein distribution. LBP plays a key role in endotoxin signaling and binds to LPS to stimulate cell activation. As a glycolipid located in the outer membrane of Gram-negative bacteria, LPS can potently induce protective and potentially harmful inflammatory responses. Increased plasma endotoxin activity in AIS patients is associated with unfavorable short-term functional outcomes (26). Changes in commensal and pathogenic microorganisms may induce intestinal mucosal barrier disruption (27), which may contribute to brain infarction exacerbation.

Here, we further identified 25 OTUs in the Ruminococcaceae family that were closely related to mouse outcome indices, namely, 22 OTUs that were negatively associated with worse brain injury or gut barrier dysfunction and three OTUs positively associated with these indices. Both OTUs in *Lachnospira* were negatively associated with impaired brain injury and gut barrier dysfunction. Ruminococcaceae and *Lachnospira* are both SCFA-producing bacteria that orchestrate the epithelial barrier to maintain gut integrity (28, 29), restore gut barrier function (30, 31), and prevent translocation and dissemination of gut bacteria and related components. In addition to SCFAs, alterations in gut microbial composition, such as increased abundance of LPS-producing bacteria, can also affect gut barrier integrity (32). As expected, *S24-7* and Rikenellaceae from LPS-producing bacterial families showed opposite correlations compared with those of the SCFA-producing bacteria. S24-7 contributed to cerebral cavernous malformation lesion formation through the LPS-TLR4 pathway associated with disruption of the gut barrier (33) and showed a positive correlation with immune responses, including NF-$\kappa$B activity (34). Decreased abundance of Ruminococcaceae and increased Rikenellaceae were observed in high-fat diet mice (35). Here, we provide relationships between the gut microbiota and brain infarction and gut barrier dysfunction indices. There were four mice in the T2D-treated GF mouse group and five in the control group. The number of GF mice was rather limited. Future studies that define the causative relationship between SCFA- or LPS-producing bacteria and brain injury poststroke are required.

In the analysis of predicted metabolic profiles, we found that functional pathways involved in energy metabolism and oxidative stress were significantly elevated in T2D-treated mice compared with control-treated GF mice. Our results indicated that the energy metabolism pathway may be influenced by gut microbial dysbiosis in T2D patients and correlated with exacerbated brain injury poststroke. Evidence has shown that the gut microbiota may affect the lipid metabolism process of the host and is associated with many risk factors for stroke, including obesity (36), diabetes (11, 24) and atherosclerosis (37). Our results showed a short-term effect of gut microbial metabolism dysbiosis on brain injury poststroke. Another study showed that lipid metabolism may exacerbate stroke outcome by developing depression poststroke. Therefore, exacerbated brain injury and gut barrier dysfunction in T2D-treated GF mice may be mediated by the gut microbiome and metabolic dysbiosis. Future studies are

needed to focus on the potential energy metabolism functions and their interactions with stroke outcome.

## MATERIALS AND METHODS

**Clinical study design and sample collection.** The human observational study was approved by the Ethics Committee of Southern Medical University (NFEC-2016-148). Written informed consent was obtained from all subjects. Patients diagnosed with AIS, including AIS patients and AIS_T2D patients, were consecutively recruited if they met the following criteria: (i) presented with ischemic stroke due to cerebral infarction (according to standard clinical criteria with supporting cerebral imaging evidence consisting of either computed tomography or magnetic resonance imaging and magnetic resonance angiography); (ii) had no evidence of hemorrhagic infarction; (iii) were >18 years of age; (iv) were admitted between 2 and 7 days after stroke onset; and (v) had a NIHSS <16 on admission. The exclusion criteria included the following: (i) severe comorbidities; (ii) unstable medical condition; (iii) chronic inflammatory disease; and (iv) treatment with antibiotics during the preceding 1 month. Patients with AIS were assessed by the NIHSS on admission and mRS and Essen stroke risk scores at 2 years poststroke. NIHSS is the most widely used measure of neurologic severity caused by AIS (38). The Essen stroke risk score and mRS score are used for risk stratification and quantification of the severity and outcome in AIS patients, respectively (39, 40). The World Health Organization (WHO) criteria were used for the definition of T2D (41). The participants were consecutively recruited from Nanfang Hospital (Southern Medical University, Guangzhou, China). A total of 90 AIS patients, 60 AIS patients with T2D, and 35 T2D patients were recruited to participate in this study. The first available fecal sample after admission and the clinical data were collected from all participating patients.

The control group included 55 healthy people who resided in Guangzhou City for more than 5 years. They had no cardiovascular or cerebrovascular diseases. The exclusion criteria for the control group were the same as those described above. Fresh fecal samples and demographic information were collected for the control group. All samples were frozen at −80°C until analysis.

**16S RNA sequencing and microbiome data analysis.** In the clinical study, bacterial genomic DNA was extracted from fecal samples using the PowerSoil DNA Extraction Kit (Mo Bio) following the manufacturer's specifications (42). A total of 240 individually processed human fecal gDNA extractions were PCR amplified. The barcoded primers 514F (GTGCCAGCMGCCGCGGTAA) and 805R (GGACTACHVGGGTWTCTAAT) were used to amplify the 16S rRNA gene V4 variable region. The PCR cycle conditions were as follows: 94°C for 2 min; followed by 30 cycles at 94°C for 30 s, 54°C for 30 s, and 72°C for 45 s; and a final elongation at 72°C for 5 min (43). All PCR amplicons were mixed together and sequenced using Illumina paired-end sequencing following the manufacturer's protocol. The raw sequences were preprocessed and quality controlled using QIIME 2 with default parameters (44) and then demultiplexed and clustered into species-level OTUs with 97% similarity. OTU generation was based on the USEARCH algorithm (45).

In animal experiments, fecal samples of GF mice after they received FMT were collected. DNA extraction was performed according to the QIAamp minikit manufacturer's instructions. Amplification and sequencing of bacterial 16S rRNA genes were performed by the Illumina ISEQ 100 platform following the manufacturer's protocol. The raw sequences were denoised with DADA2 (1.6.0) to generate amplicon sequence variants (ASVs).

Representative sequences were determined because of sequence frequency, classified by Greengenes v13_8 and aligned using the PyNAST algorithms (46). According to the representative sequence alignment, we used Fast-Tree to determine the phylogenetic relationships (47). UniFrac distances were used to analyze the $\beta$-diversity by illustrating the phylogenetic dissimilarity among samples. PCoA is a dimensionality reduction method that describes the relationship among samples based on the distance matrix and visualizes the unsupervised grouping pattern of the microbiome. The Adonis test was performed using QIIME 2 software to analyze the multidimensional microbiome data. LEfSe was used to compare the discriminative data between groups (48). LEfSe determines the abundant features most different between the conditions in accordance with biologically meaningful categories by emphasizing statistical significance, biological consistency, and effect relevance. For each differential feature in LEfSe analysis, we calculated a linear discriminant analysis value to represent the difference in the feature between groups. KEGG functions in GF mice were predicted by using PICRUSt. The differences in KEGG pathways between T2D-treated GF mice and control-treated GF mice were determined by using statistical analysis of metagenomic profiles (STAMP) (49).

**T2D gut microbial dysbiosis index.** We formulated a T2D index to represent the overall gut dysbiosis associated with T2D. The index is calculated as

$$T2D\ index = \sum \left[ (\log_{10} P\ value) \times abundance(T2D\ enriched) \right]$$
$$- \sum \left[ (\log_{10} P\ value) \times abundance(control\ enriched) \right]$$

We filtered the bacterial genera that exhibited a prevalence lower than 10%. The Wilcoxon rank sum test was used between T2D patients and healthy controls, and 11 significantly different genera were selected ($P < 0.05$), including Porphyromonas, Lactobacillus, Blautia, Dorea, Lachnospira, Roseburia, Megasphaera, Catenibacterium, Cetobacterium, and two genera from Enterobacteriaceae. This formula was used to calculate the T2D gut microbial dysbiosis index.

**Animal experimental design. GF mice.** The GF mouse experiment was performed in accordance with the institutional guidelines and regulations and was approved by the Ethics Committee of Jinan University (IACUC-20180409-02). GF male ICR mice were raised in the Experimental Animal Research Center at

**TABLE 3** Primers used in quantitative real-time PCR in this study

| Organism | Target gene | Sequence |
|---|---|---|
| *Mus musculus* | *Tjp1* | 5′-CTCCAGGTGCTTCTCTTGCT-3′ |
| | | 5′-TATCTTCGGGTGGCTTCACT-3′ |
| *Mus musculus* | *Ocln* | 5′-ACGGACCCTGACCACTATGA-3′ |
| | | 5′-TCAGCAGCAGCCATGTACTC-3′ |
| *Mus musculus* | *Cldn4* | 5′-ATCGTTGTCCGCGAGTTCTA-3′ |
| | | 5′-GCTTGTCGTTGCTACGAGGT-3′ |
| *Mus musculus* | *Cxcl1* | 5′-TGCACCCAAACCGAAGTCAT-3′ |
| | | 5′-TTGTCAGAAGCCAGCGTTCAC-3′ |
| *Mus musculus* | *Cxcl2* | 5′-TCCAGGTCAGTTAGCCTTGC-3′ |
| | | 5′-CGGTCAAAAAGTTTGCCTTG-3′ |
| *Mus musculus* | *Actb* | 5′-AGAGGGAAATCGTGCGTGAC-3′ |
| | | 5′-CAATAGTGATGACCTGGCCGT-3′ |

Jinan University. Throughout the experiment, they were given the same standard chow and autoclaved water *ad libitum* and housed in a controlled environment (23°C; inverted 12-h daylight cycle; 40% humidity). The general health status of the mice was checked by investigators every day. All mice were 8 weeks old and were comparable to normal health before the intervention. This experiment was performed in sterile environment.

**FMT.** Fecal samples were frozen at −80°C until they were chosen and used in FMT, and all samples were handled under anaerobic conditions. Samples of T2D participants (*n* = 3, male) and healthy controls (*n* = 3, male) were randomly chosen. Each fecal sample (0.1 g) was mixed with 1.5 mL of sterile physiological saline, and pools were made from equal volumes of the donor suspensions. The suspensions were mixed and centrifuged (2,000 rpm, 10 min), and the supernatants were collected. Then, 200 $\mu$L of supernatant was administered to each mouse by gastric gavage. The GF mice were randomized into two groups, which were transplanted with fecal samples from T2D patients or healthy controls every other day for 1 week. Two groups of mice were bred separately in different cages to prevent normalization of their gut microbiota.

**MCAO, evaluation of brain infarct sizes, and mouse neurological deficits.** MCAO and evaluation of brain infarct lesions were performed as described by Xu et al. (8). GF mice were occluded for 60 min. The overall mortality rate during this procedure was 10%. At 24 h after MCAO, neurological deficits were evaluated by the mNSS, in which the severity of injury was graded on a scale of 0 to 14 (50). All evaluation processes were conducted by two investigators who were blinded to the experimental groups. Then, the mice were anaesthetized, and samples were collected for further analysis.

**Blood samples.** Blood samples of GF mice were collected from the orbital plexus vein and centrifuged to obtain serum for biochemical analyses. Mouse serum LPS and LBP levels were determined using ELISA kits (Elisalab) according to the manufacturer's protocol.

**Quantitative determination of mRNA levels in intestinal tissue.** The relative transcription levels of mRNAs for ZO-1, occludin, claudin-4, chemokine (C-X-C motif) ligand 1, and chemokine (C-X-C motif) ligand 2 encoded by the *Tjp1*, *Ocln*, *Cldn4*, *Cxcl1*, and *Cxcl2* genes, respectively, were determined by qRT-PCR. Ileum and colon tissues acquired from GF mice were homogenized using a Mini Beadbeater (TIANGEN), and RNA was extracted by the TRIzol reagent method (Thermo Fisher Scientific). cDNA was generated with TaKaRa reverse transcription reagents. Real-time PCR was performed by using a ViiA 7 real-time PCR system (TaKaRa). We analyzed the data with the comparative $C_t$ method. Target gene transcription of each sample was normalized to the respective levels of $\beta$-actin mRNA. The primers used in this study are listed in Table 3.

**Immunofluorescence.** GF mouse ileum and colon tissues were removed, washed with PBS, and fixed in 4% paraformaldehyde, dehydrated, cleared, and embedded in paraffin. The paraffin sections were deparaffinized, rehydrated, and treated with EDTA antigen retrieval buffers. The slides were immunostained with anti-ZO-1 (1:100, Proteintech Group) and anti-occludin (1:50, Proteintech Group) antibodies overnight at 4°C, followed by incubation with a goat anti-rabbit Cy3-conjugated secondary antibody (1:200, Servicebio) for 1 h in total darkness. DAPI was used for nuclei staining. Fluorescence images were captured using a Nikon Eclipse Ti-SR fluorescence microscope at a magnification of ×20.

**Statistical analysis.** Statistical analysis was implemented using R3.5.1 and SPSS 20.0 (IBM, USA). The mean ± SEM and percentages were used to express continuous and discrete data. The Wilcoxon rank sum test was used to determine significant differences between different groups. Bonferroni adjustment was used for multiple comparisons. Pearson's chi-square test was used to determine the significant differences in categorical data between two groups. Correlation analysis was calculated by the Spearman correlation test. $P < 0.05$ was considered statistically significant.

**Data availability.** The raw data for 16S rRNA gene sequences in this study are available from the Sequence Read Archive (https://www.ncbi.nlm.nih.gov/bioproject) at accession number PRJNA743790.

## SUPPLEMENTAL MATERIAL

Supplemental material is available online only.

**FIG S1**, TIF file, 17.2 MB.

**FIG S2**, TIF file, 25.4 MB.

## ACKNOWLEDGMENTS

We gratefully acknowledge all the workers from the Experimental Animal Research Center at Jinan University for their support during the GF mouse experiment.

K.X. and W.S. conceived the study and designed the research. K.X., X.C., Q.W., J.Z., and G.X. performed the sampling, collected clinical patient data, processed samples, and conducted the experiments. K.X. and Y.H. analyzed the data. X.G. and H.W. performed the animal experiments. K.X., W.S., and Y.H. wrote the manuscript and revised the article. All authors have read and approved the final version of the manuscript.

This study was funded by the National Natural Science Foundation of China (NSFC31800415, NSFC82022044, NSFC81800746, and NSFC32171155), China Postdoctoral Science Foundation (2018M630967), and Science and Technology Program of Guangzhou, China (201904010091).

We declare no conflict of interest.

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
