## [Reviewer comments · mSystems]

Gut microbial dysbiosis associated with type 2 diabetes aggravates acute ischemic stroke

Xiaojiao Chen, Qiheng Wu, Xuxuan Gao, Huidi Wang, Jiajia Zhu, Genghong Xia, Yan He, Wei Song, and Kaiyu Xu

Corresponding Author(s): Kaiyu Xu, Southern Medical University

Review Timeline:

Submission Date:

November 2, 2021

Accepted:

December 6, 2021

Editor: Joshua Elias

Reviewer(s): Disclosure of reviewer identity is with reference to reviewer comments included in decision letter(s). The following individuals involved in review of your submission have agreed to reveal their identity: Xiaoquan Su (Reviewer #1)

Transaction Report:

DOI: <https://doi.org/10.1128/mSystems.01304-21>

December 6, 2021

Dr. Kaiyu Xu
Southern Medical University
Guangzhou
China

Re: mSystems01304-21 (Gut microbial dysbiosis associated with type 2 diabetes aggravates acute ischemic stroke)

Dear Dr. Kaiyu Xu:

Your manuscript has been accepted, and I am forwarding it to the ASM Journals Department for publication. For your reference, ASM Journals' address is given below. Before it can be scheduled for publication, your manuscript will be checked by the mSystems senior production editor, Ellie Ghatineh, to make sure that all elements meet the technical requirements for publication. She will contact you if anything needs to be revised before copyediting and production can begin. Otherwise, you will be notified when your proofs are ready to be viewed.

Publication Fees:

We recognize that the video files can become quite large, and so to avoid quality loss ASM suggests sending the video file via <https://www.wetransfer.com/>. When you have a final version of the video and the still ready to share, please send it to Ellie Ghatineh at eghatineh@asmusa.org.

Sincerely,

Joshua Elias
Editor, mSystems

Journals Department
FIG S2: Accept
FIG S1: Accept